

# First record of *Apanteles hemara* (N.) on *Leucinodes orbonalis* Guenée and biodiversity of Hymenoptera parasitoids on Brinjal

Hager M. M. Saleh[1,2], Areej A. Al-Khalaf[3], Maha Abdullah Alwaili[3] and Debjani Dey[1]

[1] National Pusa Collection, Division of Entomology, ICAR-Indian Agricultural Research Institute, New Delhi, India
[2] Plant Protection Department, Faculty of Agriculture, Fayoum University, Fayoum, Egypt
[3] Biology Department, Faculty of Science, Princess Nourah Bint Abdulrahman University, Riyadh, Saudi Arabia

## ABSTRACT

The brinjal fruit and shoot borer (BFSB), *Leucinodes orbonalis* Guenée (Lepidoptera: Crambidae), is a very detrimental pest that causes significant economic losses to brinjal crop worldwide. Infested brinjal fruits were collected from vegetable fields located at the ICAR-Indian Agricultural Research Institute (ICAR-IARI), New Delhi, India, during two consecutive seasons (2021–2022). The larvae of the pest were brought to the laboratory and reared under controlled conditions of 25 ± 0.5 °C and 70 ± 5% relative humidity, for the emergence of parasitoids. In addition, the survey of Hymenoptera parasitoids in brinjal was conducted utilizing a sweep net and yellow pan trap over the course of two seasons. The results reveal that five parasitoid species were emerged from *L. orbonalis* viz., *Apanteles hemara* Nixon, 1965, *Bracon greeni* Ashmead 1896 (Hymenoptera: Braconidae), *Goryphus nursei* (Cameron, 1907), *Trathala flavoorbitalis* (Cameron, 1907) (Hymenoptera: Ichneumonidae) and *Spalangia gemina* Boucek 1963 (Hymenoptera: Spalangiidae). Out of these, *A. hemara* and *S. gemina* were documented as new occurrences in Delhi. Additionally, *A. hemara* was recorded for the first time as a parasite on *L. orbonalis*. *Trathala flavoorbitalis* was observed during both seasons and exhibited higher parasitism reaching 15.55% and 18.46% in July and August 2022, respectively. However, the average parasitism (%) recorded by *A. hemara*, *B. greeni*, *G. nursei*, *T. flavoorbitalis* and *S. gemina* was 3.10%, 1.76%, 1.10%, 9.28% and 1.20% respectively. Furthermore, the findings showed a significant ($p \leq 0.01$) strongly positive correlation between fruit infestation (%) by *L. orbonalis* and parasitism (%). The survey indicates the presence of a broad group (19 families and 60 species) of Hymenoptera parasitoids in the brinjal crop ecosystem in Delhi which could be valuable in biological control. In light of these results, this study revealed that *A. hemara* and other parasitoids identified in this study alongside *T. flavoorbitalis* would be ideal biocontrol agents within the integrated pest management (IPM) program of BFSB in Delhi.

Corresponding authors
Hager M. M. Saleh,
hms03@fayoum.edu.eg
Debjani Dey, ddeyiari@hotmail.com

# INTRODUCTION

Brinjal (*Solanum melongena* L.) occupies a notable position as the third most extensively consumed and economically valued vegetable in Asia (*FAOSTAT, 2023*). India is the second largest producer of brinjal in the world, next to China. Despite the overall growth in global brinjal production, its productivity is constrained by challenges posed by insects and diseases (*Alam & Salimullah, 2021*). The predominant insect pest that inflicts damage to brinjal crop is the lepidopteran fruit and shoot borer, which seriously threatens its cultivation (*Mainali, 2014*).

The Brinjal fruit and shoot borer (BFSB) *Leucinodes orbonalis* Guenée (Lepidoptera: Crambidae) is a highly important insect pest that affects brinjal plants worldwide (*Nusra et al., 2021*). The larvae of BFSB cause damage to eggplant plants by tunneling into the leaf petioles, midribs, and tender shoots, which subsequently leads to the wilting and desiccation of the stems. Furthermore, the larvae feed on the flowers, causing them to fall prematurely or resulting in malformed fruits (*Shelton et al., 2019*).

The most significant economic damage caused by BFSB occurs to the fruit itself. The fruit becomes unsuitable for human consumption and sale due to the presence of holes, feeding tunnels, and excrement from larvae. BFSB has a high ability to reproduce, quickly cycles through generations, and causes extensive damage in both wet and dry seasons, posing a significant challenge to plants (*Prodhan et al., 2018*).

Infestation levels may exceed 90%, resulting in substantial worldwide economic losses estimated at 86–90% (*Ghosh, Laskar & Senapati, 2003*). In India, brinjal is sprayed with chemicals 15–40 times per season (*Watkins et al., 2012*). However, this approach is environmentally hazardous, poses potential health risks to both consumers and farmers, and incurs significant costs (*Prodhan et al., 2018*).

To adopt an environmentally friendly strategy for controlling pests, it is imperative to protect and preserve the natural predators and parasitoids that naturally keep pest populations in check. Among the 21 parasitoids reported in relation to BSFB, one of the most prominent parasitoids is *Trathala flavoorbitalis* (Cameron, 1907) (Hymenoptera: Ichneumonidae) (*Ranjith et al., 2020*). This parasitoid exhibits a notable parasitism rate of 61.70% (*Srinivasan, 2008*). In addition to *T. flavoorbitalis*, *Goryphus nursei* (Cameron, 1907) (Hymenoptera: Ichneumonidae) was recorded in Uttar Pradesh and proved to be an effective parasitoid, displayed a maximum parasitism rate of 7% particularly in the winter season (*Alam et al., 2003*).

*Trathala flavoorbitalis*, a widely recognized parasitoid of *L. orbonalis*, has been observed in different regions of India, especially Bihar, Tamil Nadu, Manipur and Karnataka (*Mallik et al., 1989*; *Yasodha & Natarajan, 2009*; *Murali et al., 2017*; *Ranjith et al., 2020*; *Thokchom et al., 2022*). This parasitoid species has a cosmopolitan distribution, and is known from the Afrotropical, Autralasian, Eastern Palaearctic, Indomalayan, Nearctic, and Oceanic regions (*Rousse & Villemant, 2012*).

In Tamil Nadu, *Yasodha & Natarajan (2009)* documented the emergence of 12 parasitoid species from BFSB, belonging to the superfamilies Ichneumonoidea and Chalcidoidea. Furthermore, *Murali et al. (2017)* reported the presence of *Spalangia gemina* (Hymenoptera: Spalangiidae) on BSFB, while *Bracon greeni* (Hymenoptera: Braconidae) was documented by *Venkatraman, Gupta & Negi (1948)*.

Consequently, this research aims to evaluate the parasitism rate of parasites associated with BFSB throughout two seasons, which is crucial for the success of its biological control. Furthermore, we present a survey of Hymenoptera parasitoids families in the brinjal crop ecosystem in New Delhi.

## MATERIALS AND METHODS

### Insect sampling

This study was carried out to record and assess the potential of the natural enemies associated with the *L. orbonalis* Guenée during two consecutive seasons (November 2021–October 2022) in vegetable fields located at ICAR-Indian Agricultural Research Institute (ICAR-IARI), New Delhi, India.

The weather parameters during the study period such as rainfall, minimum and maximum temperature and relative humidity were recorded from an agrometeorological observatory, Division of Agricultural Physics, IARI, New Delhi and are provided in Table S1.

To calculate the pest infestation rate (%) in brinjal, the field was divided into four quarters, and 15 plants per quarter were randomly checked. The pest incidence was observed at 7 days' intervals, and infestation (%) was calculated according to the following equation.

$$\textbf{Fruit infestation } (\%) = \frac{\text{No. of infested fruits}}{\text{Total fruits observed}} \times 100. \qquad (1)$$

Infested brinjal fruits were collected, and the larvae were reared for the emergence of parasitoids under laboratory conditions of 25 ± 0.5 °C, relative humidity 65 ± 5%, and a photoperiod of 12 light: 12 Dark h. The larvae, after pupation were separated and kept in plastic containers until parasitoid emergence. The parasitoids that emerged were preserved in 70% alcohol and card mounted for taxonomic studies. The parasitism percentage during each month was calculated according to the following equation (*Van Driesche, 1983*).

$$\textbf{Parasitism } (\%) = \frac{\text{Total parasitoids emerged from larvae or pupa}}{\text{Total No. of larvae or pupa collected from field}} \times 100. \qquad (2)$$

In addition, a randomized weekly survey was carried out throughout two seasons to study the presence of Hymenoptera parasitoids in brinjal utilizing a sweep net (consists of a net with a 35 cm diameter and 40 cm length held on a circular aluminum frame which is connected to aluminum handle of 80 cm in length) and yellow pan trap (consists of a shallow yellow tray, about 5 cm deep and 15 cm diameter).

## Identification of parasitoids

From the collected specimens, large parasitoids were pinned and dry-preserved, while small ones were preserved in 70% ethanol. Parasitoids were identified with the help of *Bouček (1951)*, *Habu (1960)*, *Stary (1975)*, *Husain & Agarwal (1982)*, *Greco (1997)*, *Narendran, Galande & Mote (2001)*, *Belokobylskij (2003)*, *Jonathan (2006)*, *Gibson (2009)*, *Rousse, Villemant & Seyrig (2011)*, *Rousse & Villemant (2012)*, *Sheeba & Narendran (2013)*, *Xu, Olmi & He (2013)*, *Akhtar, Singh & Ramamurthy (2014)*, *Ghafouri-Moghaddam, Lotfalizadeh & Rakhshani (2014)*, *Amer, Zeya & Veenakumari (2016)*, *Cao, LaSalle & Zhu (2017)*, *Fernandez-Triana et al. (2017)*, *Khalaim (2018)*, *Ahmed et al. (2020)*, *Zerova & Fursov (2020)*, *Gull-e-Fareen et al. (2021)*, *Talamas et al. (2021)*, *Burks et al. (2022a)*. The morphological studies were carried out using a Leica S8AP0 stereo microscope and a LEICA M205 C stereozoom automountage microscope. Multi-focused montage images were taken using a LEICA MC190 HD digital camera attached to the LEICA M205 C stereozoom automountage microscope. The photographs and illustrations were processed with Adobe Photoshop CS5 and plates were then prepared. The morphological terminology and wing venation are based on *Nixon (1965)*, *Jonathan (2006)*, *Gibson (2009)*, *Sheeba & Narendran (2013)*. All the specimens have been deposited in the National Pusa Collection (NPC), ICAR-IARI, New Delhi, India.

## Statistical analysis

The parasitism (%) was analyzed statistically, and the extent of parasitism (%) was also subjected to correlation analysis with brinjal fruit infestation (%) as well as with weather factors using Minitab® Statistical Software (17.0).

## RESULTS

The findings of this study revealed the presence of five parasitoid species recorded on *L. orbonalis*: *A. hemara*, *B. greeni*, *G. nursei*, *T. flavoorbitalis* and *S. gemina*.

### Systematic study

The main diagnostic characteristics of five parasitic wasps that emerged from Brinjal fruit and shoot borer (*L. orbonalis* Guenée), as well as their hosts and distribution details, were highlighted as follows.

### *Apanteles hemara Nixon, 1965* (Hymenoptera: Braconidae) (Figs. 1A–1D)

**Diagnosis**: Body length 3 mm, Body colour black. Mandible brown, labrum orange testaceous, palpi pale yellow, clypeus dark brown and antennae black. Fore and mid legs yellowish orange, entire hind coxa densely punctate rugose; hind femur brown, yellow trochanter, hind tibia most often strikingly bicolor, yellow and brown on posterior with spurs white, hind tarsus infuscate, ovipositor yellow, ovipositor sheet dark brown. Wing hyaline, venation brown; pterostigma mostly brown. Head entirely densely setose. Clypeus ventral margin slightly concave. Face and clypeus moderately and shallowly punctate; antenna slightly shorter than body length, with 16 segments; OOL 1.5x POL. Mesosoma,

mesoscutum and scutellum with relatively coarse and dense punctures (distance between punctures smaller than diameter of puncture). Propodeum with areola complete, propodeal areola strong, centrally smooth, apically pointed and basally petiolate to base of propodeum by two sub-median irregular carinas (Fig. 1D). Forewing with 2Rs more than twice shorter than r, Vein r length 1.7x vein 2RS, 1-R long, Vein 1-R length 1.2x pterostigma length. The areolet open (Fig. 1B). Legs hind coxae entirely punctate rugose. Hind femur slightly swollen. Metasoma T1 of metasoma much longer than wide with strong, longitudinal striation, its margins sub-parallel to strongly converging apically. Tergum 2 wider than long, short, transverse and apically widened. Tergum 3 longer than tergum 2 (Fig. 1C). Ovipositor sheath slightly shorter than tibia. Ovipositor large, usually slightly decurved and gradually tapered.

**Material examined:** 1♀ June; 2♀ July; 2♀, 1♂ August; 1♀ September 2022 emerged from *L. orbonalis* and 1♀ August; 2♀ September 2022 IARI, New Delhi, yellow pan trap on brinjal (HC).

**Host records**: *Tebenna micalis* (Choreutidae); *Cnaphalocrocis trapezalis*, *Herpetogramma stultalis*, *Hydriris ornatalis*, *Omiodes indicatae*, *Spoladea recurvalis* and *Udea ferrugalis* (Crambidae) (*Fernandez-Triana et al., 2017*), *Hymenia fascialis*, *Pachyzancla stultalis* (Pyralidae) (*Nixon, 1965*).

**Distribution**: Kenya, Madagascar, Cape Verde Islands, Mauritius, Saudi Arabia, Senegal, Republic of Congo, South Africa, Yemen, Australia, and India (new record established at New Delhi) (*Fernandez-Triana et al., 2017*).

### *Bracon greeni* Ashmead 1896 (Hymenoptera: Braconidae) (Figs. 2A–2E)

**Diagnosis**: Body length 3.1 mm, body colour brownish-yellow. Disc of metasoma, extreme apex of second tergite and large dorsal blotches on third and fourth tergites black; wings hyaline, stigma and veins brown, ocellar triangle black (Figs. 2A, 2C). Head smooth, wider than long; antenna 24 segments, nearly as long as the body length. vertex rugose, anteriorly smooth, shiny posteriorly with setose; eyes glabrous, length of eye in dorsal view 2.2x temple; temple smooth, shiny; width of face 1.7x its height; clypeus smooth, intertentorial distance 1.8x tentorio-ocular distance (Fig. 2B). Mesosoma: pronotum and mesoscutum smooth, shiny with notauli weakly impressed, scutellum smooth, shiny; metanotum with anterior median carina; propleuron smooth, shiny with setose; propodeum smooth, shiny with setose laterally and posterior median longitudinal carina extending upto its middle; propodeal spiracle round, small and medially placed; fore wing vein 3-SR about 3.0x vein r (Fig. 2E). Metasoma oval and shagreen; second to fourth metasomal tergites subequal, remaining a little shorter. Large dorsal blotches on third and fourth tergites black; ovipositor nearly as long as metasoma (Fig. 2D).

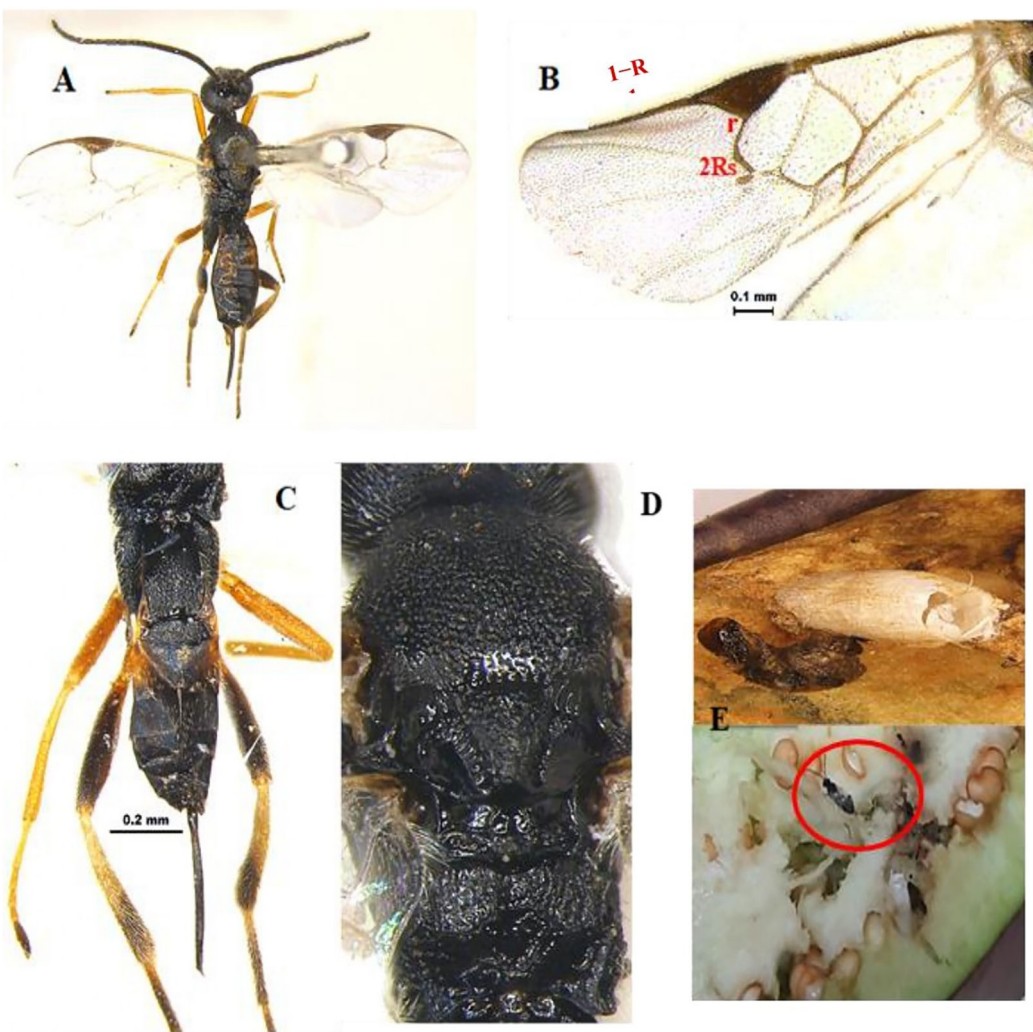

**Figure 1** *Apanteles hemara Nixon, 1965* **(female).** (A) Dorsal habitus, (B) fore and hind wings, (C) dorsal view of mesosoma, (D) dorsal view of metasoma, and (E) emerged parasite from host.

**Material examined**: 2♀ November; 1♀ December 2021; 1♀ February 2022 emerged from *L. orbonalis* and 1♀ November; 2♀ December 2021 ICAR-IARI, New Delhi, yellow pan trap on brinjal (HC).

**Host records**: *Adisura atkinsoni, Alcides affaber, Earias fabia, Heliothis obsolete, Rabila frontalis* (*Sheeba & Narendran, 2013*) and *L. orbonalis* (*Venkatraman, Gupta & Negi, 1948*). In the present study, it emerged from *L. orbonalis*.

**Distribution**: India (Kerala, Uttar Pradesh), Bangladesh, China, and Sri Lanka.

## *Goryphus nursei* (Cameron, 1907) (Hymenoptera: Ichneumonidae) (Figs. 3A–3F)

**Diagnosis**: Body length 8.2–10 mm, body colour dark orange except eighth to tenth flagellar segments, face and frons along the eye margin, apex of metasoma T1 and T5–T8

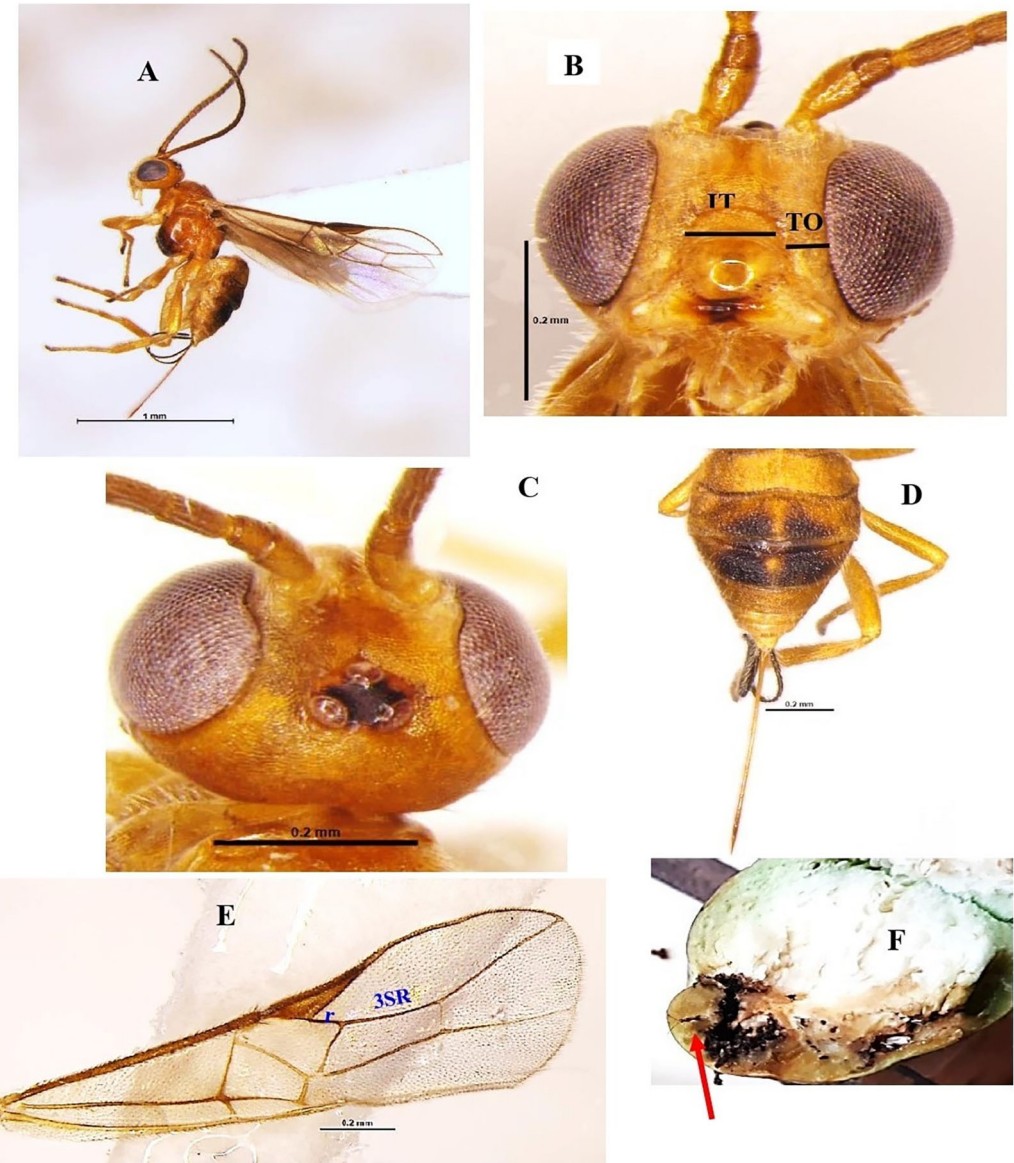

**Figure 2** *Bracon greeni* **Ashmead 1896 (female).** (A) Lateral habitus, (B) frontal view of head (TO tenterio-ocular distance; IT intertentorial distance), (C) dorsal view of head, (D) dorsal view of metasoma, (E) fore wing and (F) emerged parasite from host.

light yellow to white; T2–T4 entirely black. Legs in general reddish, except hind femur broadly at apex, whole of tibia and tarsus blackish; hind tibia with a subbasal white band (Figs. 3A, 3B). Head face and clypeus strongly punctate, clypeus tending to be smooth towards the apex, intertentorial distance 1.2x tentorio-ocular distance. Antennal scrobe shallow, smooth and shiny, temple broadly smooth and shiny, malar space weakly granulose. Frons below ocellar triangle weakly striate. Mandible thick with longitudinally striate with punctures in between the striae with teeth subequal, pointed, superior tooth 1.2x longest than the inferior. Mesosoma pronotum coarsely striate, Mesoscutum strongly punctate, striate along margins of each lobe. Scutellum, Metascutellum smooth and shiny

(Fig. 3D). Mesopleuron rugose, tending to be wrinkled above, Propodeum largely reticulate, basad of basal carina wrinkled, both transverse carinae strong and complete, basal carina more or less straight, apical carina evenly and strongly arched, spiracle small and oval. Areolet of fore wing small, squares, about 3.0x as high as the width of bordering veins (Fig. 3F). Metasoma T1 short, about 1.3x as long as wide at apex, 2$^{nd}$ and 3$^{rd}$ tergite closely punctate; 4$^{th}$ tergite less punctate. Ovipositor tip with distinct teeth ventrally (Fig. 3C).

**Material examined**: 1♀ December 2021; 2♀ February 2022 emerged from *L. orbonalis* and 1♀ November 2021; 1♀ January 2022, ICAR-IARI, New Delhi, yellow pan trap on brinjal (HC).

**Host records**: BFSB *L. orbonalis* (*Alam et al., 2003*).

**Distribution**: India (Bihar, Delhi, Gujarat, Haryana, Jharkhand, Maharashtra, Odisha, Punjab, Rajasthan, Tamil Nadu, Uttarakhand, and Uttar Pradesh), China and Pakistan (*Jonathan, 2006*).

### *Trathala flavoorbitalis* (Cameron, 1907) (Hymenoptera: Ichneumonidae) (Figs. 4A–4F)

**Diagnosis**: Body length 7.2 mm, body colour brownish-yellow, Antenna scape and pedicel yellowish and flagellomere brownish, vertex (except ocelli area black) and around eye light yellow; notauli often well marked with yellow, tegula and scutellum yellow; legs yellow, hind tibia slightly infuscate basally and apically; wings hyaline, pterostigma brown; metasoma orange, tergites 1–2 and basal triangle on tergite III dark brown, sheath and ovipositor dark brown (Figs. 4A–4F). Head strongly constricted behind eyes. Face densely punctate, clypeus smooth, vertex and frons granulate, center of frons smooth. Temples short and slightly rounded; malar space 0.65 times as long as basal mandible width, occipital carina complete; mandible with equal size teeth (Fig. 4C); antenna filiform longer than head and mesosoma with 35 segments; scape length 1.3x pedicel length. POL 0.6x OOL. Mesonotum densely punctate-shagreened, scutellum and metanotum more sparsely punctate. Scutellum rounded without dorsal lateral carinae. Propodeum densely punctate-shagreened dorsally than laterally. Propodeal carination complete, area basalis small but distinct (Fig. 4D). Metasoma tergite I a little longer than tergite II and twice longer than tergite III. Tergite I almost smooth, slightly longitudinally strigose at apex. Tergite II three times longer than apically wide and longitudinally striated (Fig. 4F); ovipositor shorter than abdomen.

**Material examined**: 3♀, 1♂ November; 2♀ December 2021; 1♀, 1♂ January; 2♀, 1♂ February; 3♀, 2♂ March; 2♀, 1♂ June; 4♀, 2♂ July; 6♀, 2♂ August; 3♀, 1♂ September; 1♀, 1♂ October 2022 emerged from *L. orbonalis* and 2♀ July; 3♀ August 2022, ICAR-IARI, New Delhi, sweeping net, on brinjal (HC).

**Host records**: Seventy-eight host records, all Lepidoptera (Gelechoidea, Noctuoidea, Pyraloidea, Tineoidea and Tortricoidea) (*Rousse, Villemant & Seyrig, 2011*) brinjal shoot

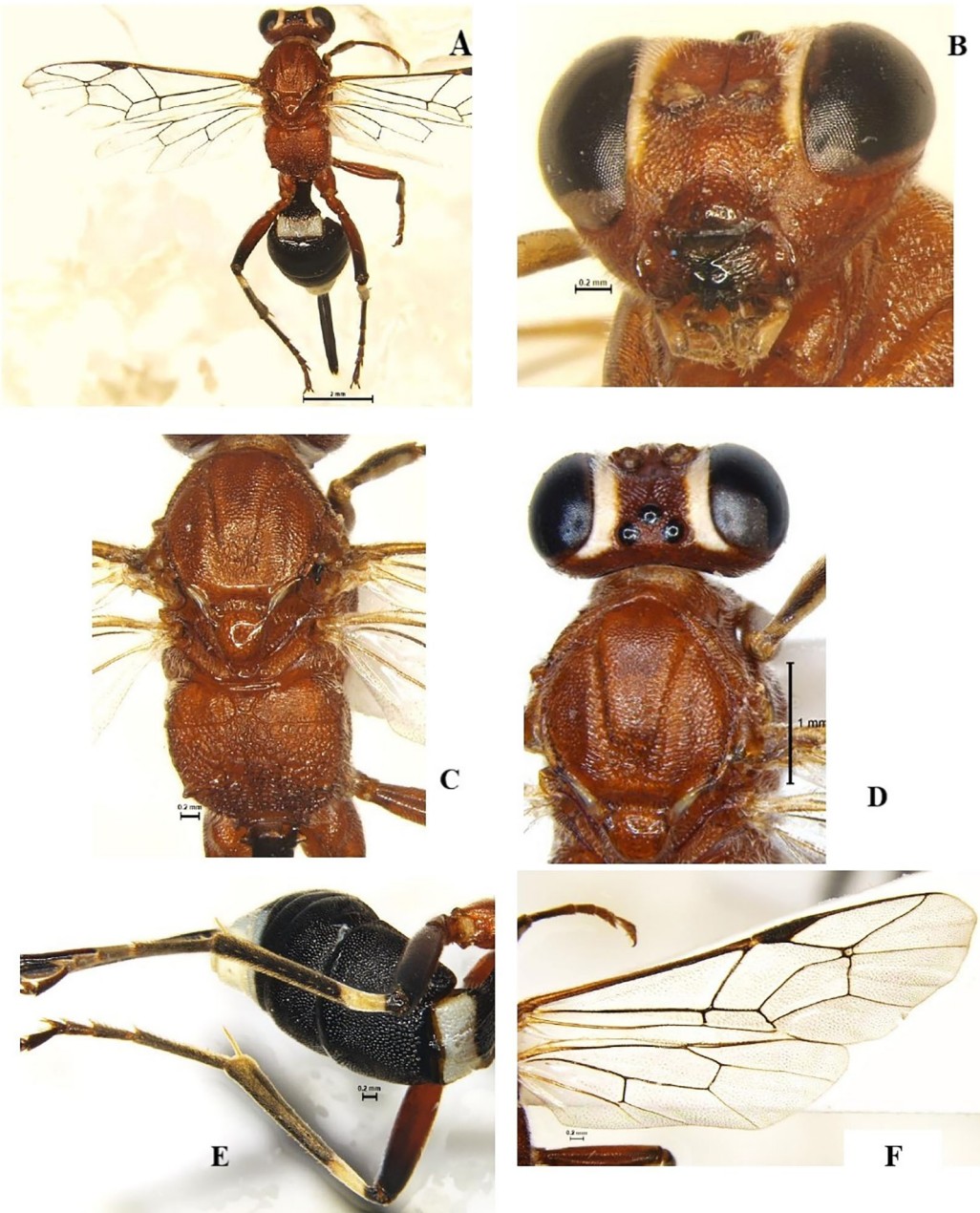

**Figure 3** *Goryphus nursei* **(Cameron, 1907) (female).** (A) Dorsal habitus, (B) frontal view of head, (C) dorsal view of mesosoma, (D) dorsal view of head, mesoscutum and scutellum, and (E) dorso-lateral view of metasoma; (F) fore and hind wings.

and fruit borer (*Alam et al., 2003*; *Ranjith et al., 2020*). In the present study, it emerged from *L. orbonalis*.

**Distribution**: Reunion, Madagascar (*Rousse, Villemant & Seyrig, 2011*). Widespread through Indo-Pacific and Eastern Oriental Areas (*Rousse & Villemant, 2012*).

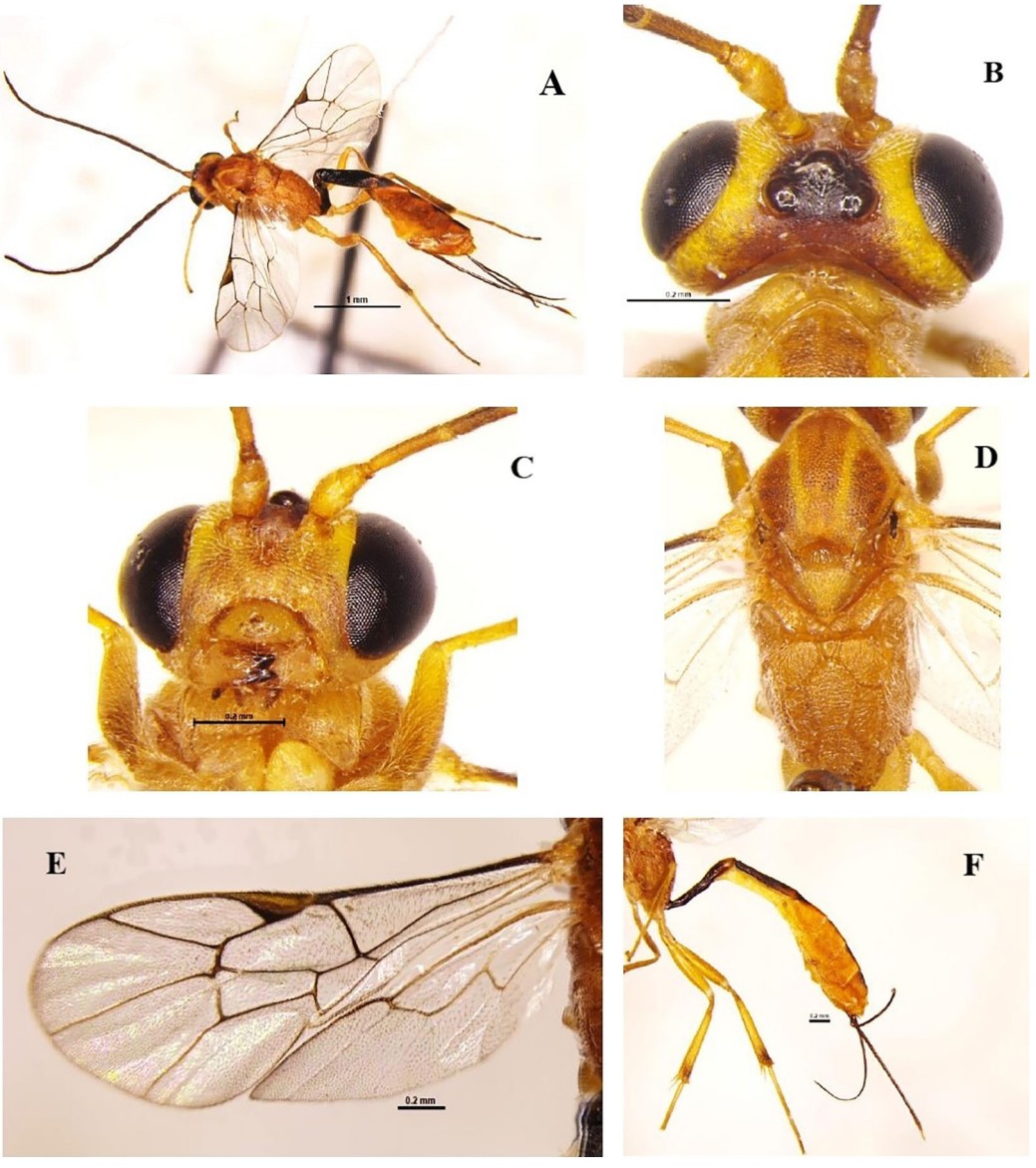

**Figure 4** *Trathala flavoorbitalis* **(Cameron, 1907) (female).** (A) Dorso-lateral habitus, (B) dorsal view of head, (C) frontal view of head, (D) dorsal view of mesosoma, (E) fore and hind wings and (F) lateral view of metasoma.

## *Spalangia gemina* Boucek, 1963 (Hymenoptera: Spalangiidae) (Figs. 5A–5E)

**Diagnosis**: Body length 3.2 mm, body colour dark. Legs dark except basal four tarsal segments yellow. Head capsule smooth and shiny with dense circular setiferous punctures; temple distinctly circular punctures. Gena with malar space distinctly shorter than eye height, scrobal depression with punctate-crenulate scrobes on either side of smooth and shiny interantennal region; the punctures separated by linear ridges and often multisided or more or less reticulate-rugose, gena densely punctate without malar sulcus, malar space about 0.8x eye height (Fig. 5B). Antenna with scape about 6.6x as long as greatest width,

the inner and outer surfaces uniformly setose and strongly. Mesosoma pronotal collar in lateral view convex behind neck, with coarsely reticulate-rugose, except for a nearly triangular area close to the crenulate cross-line posteriorly; axillae smooth and shiny except for a few pinprick-like setiferous punctures. Scutellum smooth and shiny except for a few pinprick-like setiferous punctures laterally (Fig. 5D); mesopleuron with longitudinal carinae extending from subalar area ventrally over almost all of upper mesepimeron, subalar scrobe extending posteroventrally along transepisternal line, and epistern (Fig. 5C). Fore wing hyaline; bare behind submarginal vein (Fig. 5E). Propodeum with postspiracular sulcus; dorsal surface punctate-rugose anteriorly and posteriorly, sculptured, sometimes almost smooth; supracoxal; propodeal sides smooth and shiny. Metasoma smooth and shiny. Petiole about 1.7x as long as medial width; almost smooth to finely, transversely carinate between longitudinal carinae.

**Material examined**:1♀ June; 2♀ August; 1♀ September 2022 emerged from *L. orbonalis* and 2♀ July; 1♀ August 2022, ICAR-IARI, New Delhi, sweeping net trap on brinjal (HC).

**Host records**: Micropezidae, Noctuidae and Tortricidae (Lepidoptera) as primary hosts (*Burks et al., 2022b*) recorded on *L. orbonalis* (*Murali et al., 2017*).

**Distribution**: Afrotropical, Australasian, Oriental, and Neotropical region (*Burks et al., 2022b*).

## Evaluation of parasitism extent by *T. flavoorbitalis* and other parasitoids on BFSB

During July and August 2022, *T. flavoorbitalis* showed higher parasitism of 15.55% and 18.46%, respectively. However, the average parasitism (%) of *A. hemara*, *B. greeni*, *G. nursei*, *T. flavoorbitalis* and *S. gemina* was 3.10%, 1.76%, 1.10%, 9.28% and 1.20% respectively. Throughout the study period, *T. flavoorbitalis* was the dominant parasitoid. Its parasitism in 1st season peaked at 12% in November 2021, followed by a decline in December 2021. However, it experienced subsequent increases in February and March 2022. During the 2nd season, its parasitism peaked at 18.50% in August 2022, and higher fruit infestation (%) was observed (Fig. S1). The parasitism (%) exhibited by *B. greeni* was 6.70% and 3.30% in November and December 2021, respectively; however, it was not recorded during the 2nd season. The parasitism (%) of *G. nursei* was 3.30% and 2.90% in December and January, respectively but it was not observed during the 2nd season (Fig. 6).

The higher parasitism (%) of *S. gemina* was 2.38% recorded in August 2022. *Apanteles hemara*, as a new parasitoid (Hymenoptera: Braconidae), was found to parasitizes on the larvae of BFSB. This species was recorded during the 2nd season, and its higher parasitism (%) was 6.12% and 4.76% recorded in August and September, respectively (Fig. 6). The total parasitism (%) exhibited by all parasitoids, as well as the fruit infestation (%) reach their peaks in August (Fig. 7).

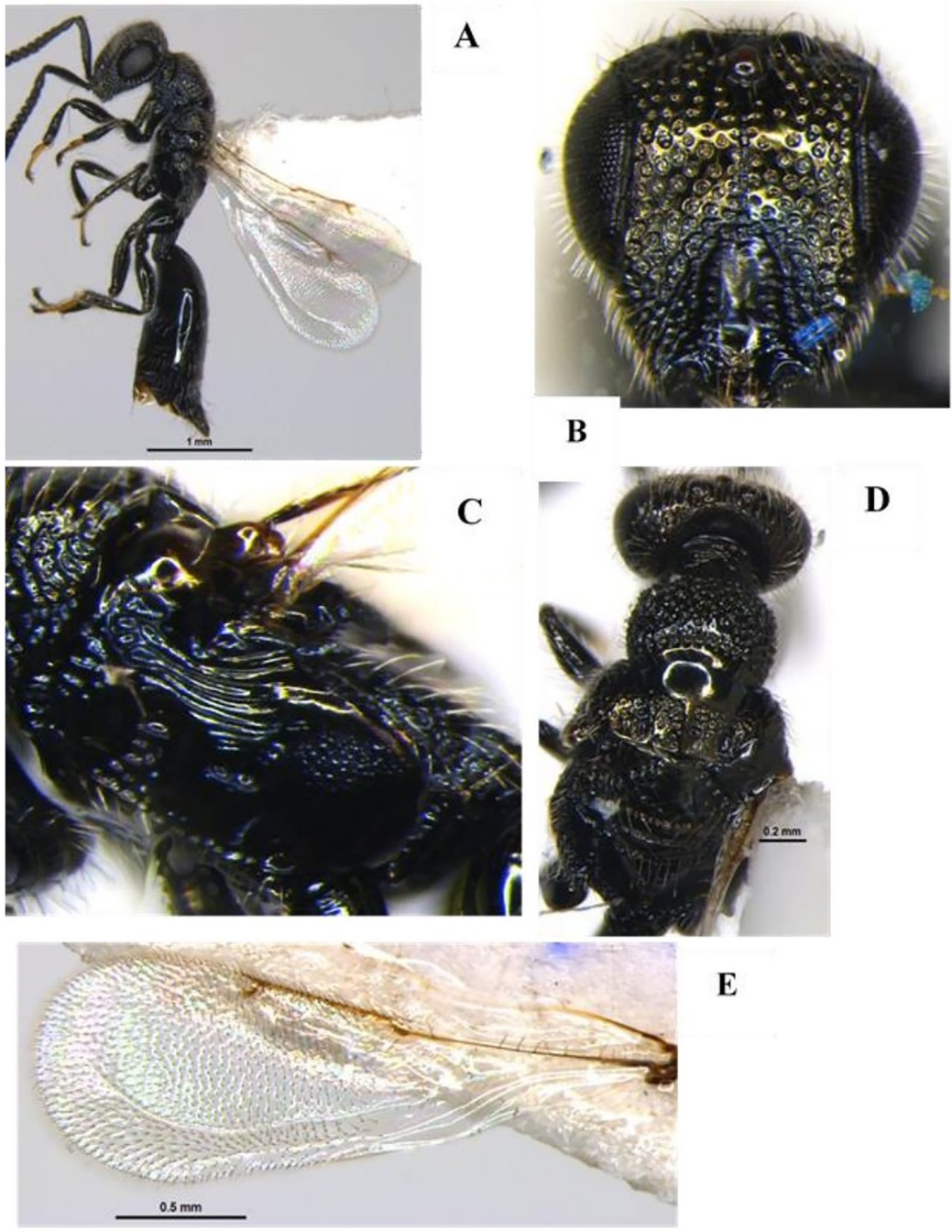

**Figure 5 *Spalangia gemina* Boucek, 1963 (female).** (A) Lateral habitus, (B) frontal view of head, (C) mesopleuron, (D) dorsal view of mesosoma and (E) fore wing.

## Correlation analysis of fruit infestation (%) and parasitism (%) under study conditions

Correlation analysis was performed to explore the interrelationship among the infestation (%) and parasitism (%) with the environmental parameters during the study period. A significant ($p \leq 0.01$) strongly positive correlation (r = 0.7) was shown between fruit

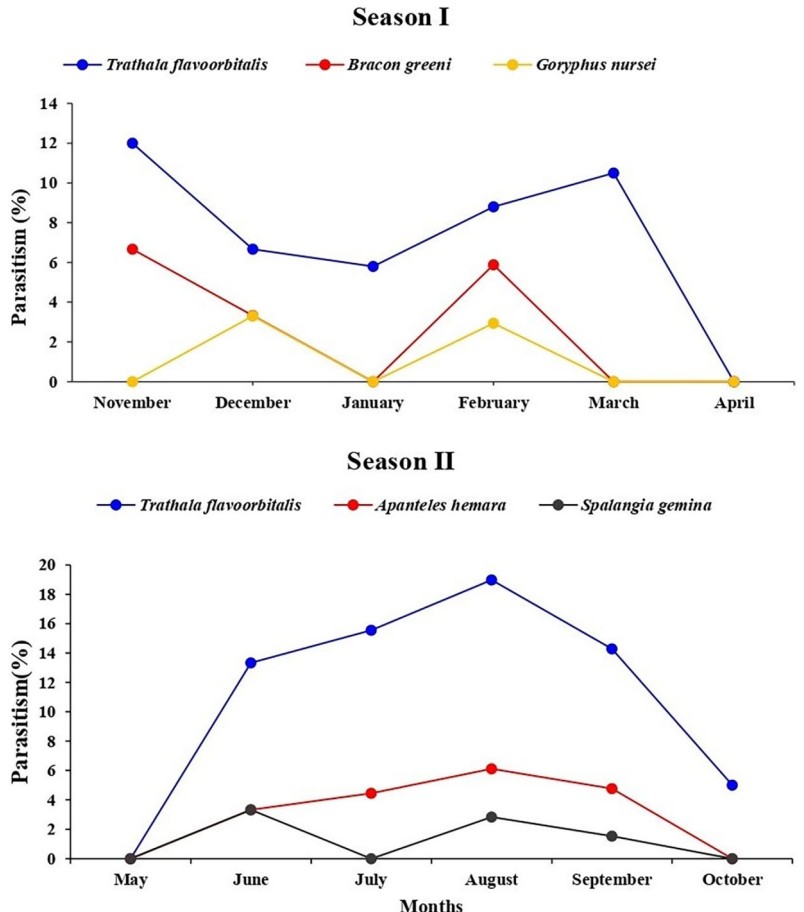

**Figure 6 Parasitism (%) of all parasitoid species on _L. orbonalis_ during two seasons (November 2021 to October 2022).**

infestation (%) and total parasitism (%) of parasitoid species. Also, a significant ($p \leq 0.01$) strongly positive correlation (r = 0.7) was recorded between infestation (%) and parasitism (%) of _T. flavoorbitalis_. Parasitism (%) of _T. flavoorbitalis_ showed a significant ($p \leq 0.5$) moderately negative correlation with rainfall (r = −0.4), moderately positive correlation with _Tmin_ (r = 0.3), RH (r = 0.4) and not correlated with _Tmax_ (r = −0.02). Fruit infestation (%) showed a significantly ($p \leq 0.1-0.5$) moderately positive correlation with _Tmax_, (r = 0.3), _Tmin_ (r = 0.5), RH (r = 0.4), and a moderate negative correlation with rainfall (r = −0.5). Total parasitism (%) showed a significant ($p \leq 0.5$) moderately negative correlation with rainfall (r = −0.3), weakly positive correlation with _Tmin_ (r = 0.2), moderately positive correlation with RH (r = 0.4) and not correlated with _Tmax_ (r = −0.05).

## Survey for study of associated Hymenoptera parasitoids

Parasitoids were collected weekly from brinjal during two seasons (November 2021–October 2022), utilizing a sweep net and yellow pan trap. The monthly distribution of different Hymenoptera parasitoid families (19 families) collected from brinjal is shown in (Fig. S2). In this study a total of 60 species were recorded, from which 48 were identified

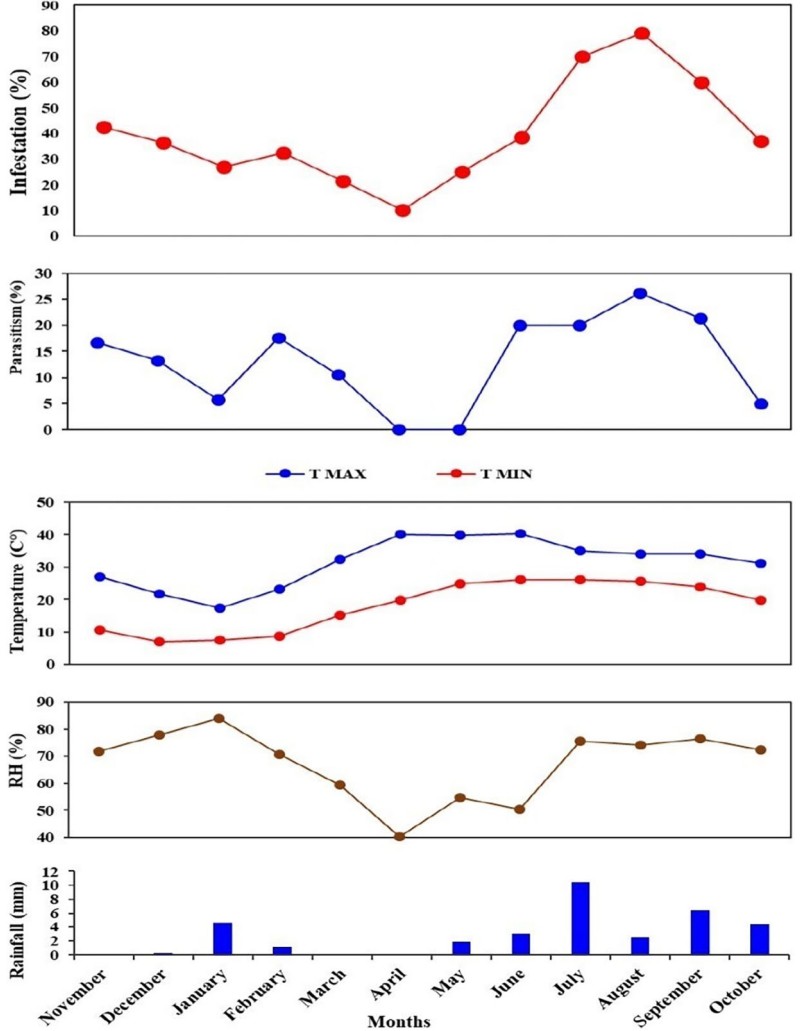

**Figure 7 Monthly fruit infestation (%) by *L. orbonalis* and parasitism (%) of parasitoids during the study period (November 2021 to October 2022).**

up to the species level, and 11 were identified up to the genus level. This provides novel insights into the distribution patterns of recently identified parasitoid species recorded on brinjal from Delhi. About 41.7% of these newly recorded species (25 out of the 60 species). The superfamily Chalcidoidea was the most dominant, followed by Ichneumonoidea, Platygastroidea and Ceraphronoidea. These 60 species belong to various families as follows: Braconidae (twelve species), Ichneumonidae (nine species), Chalcididae (seven species), Scelionidae (nine species), Pteromalidae (four species), Dryinidae (three species), two species each in the families (Diapriidae, Eulophidae, Eurytomidae, Figitidae, and Mymaridae) as well as one species was recorded in each of the families (Agonidae, Aphelinidae, Bethylidae, Ceraphronidae, Platygastridae, and Spalangiidae) (Table 1 and Figs. 8–11).

**Table 1 Surveying of the Hymenoptera parasitoids in brinjal during two seasons (November 2021 to October 2022).**

| S.No | Families/Parasitoids | 1st | 2nd | S.No | Families/Parasitoids | 1st | 2nd |
|------|----------------------|-----|-----|------|----------------------|-----|-----|
| **Superfamily: Ichneumonoidea** | | | | 36. | *Mymar taprobanicum* | + | – |
| **Family Braconidae** | | | | **Family Pteromalidae** | | | |
| 1. | *Amyosoma chinese* | – | + | 37. | *Pachyneuron solitarium*\* | + | – |
| 2. | *Apanteles hemara*\* | – | + | 38. | *Pteromalus puparum* | + | + |
| 3. | *Aphidius colemani* | + | + | 39. | *Sphegigaster brunneicornis*\* | + | + |
| 4. | *Binodoxys indicus* | + | + | 40. | *Sycoscapter* sp. | + | + |
| 5. | *Bracon carpomyiae*\* | + | + | **Family Spalangiidae** | | | |
| 6. | *Bracon greeni* | + | – | 41. | *Spalangia gemina*\* | + | + |
| 7. | *Chelonus blackburni* | – | + | **Superfamily: Cynipoidea** | | | |
| 8. | *Cotesia* sp. | + | + | **Family Figitidae** | | | |
| 9. | *Habrobracon* sp. | + | – | 42. | *Alloxysta pleuralis*\* | + | – |
| 10. | *Microcentrus delhiensis* | + | – | 43. | *Callaspidia notate*\* | + | – |
| 11. | *Phanerotoma* sp. | + | – | **Superfamily: Platygastroidea** | | | |
| 12. | *Spathius helle* | + | – | **Family Platygastridae** | | | |
| **Family Ichneumonidae** | | | | 44. | *Macroteleia livingstoni* | – | + |
| 13. | *Aneuclis* sp\* | + | – | **Family Scelionidae** | | | |
| 14. | *Diplazon laetatorius*\* | + | + | 45. | *Baryconus europaeus*\* | + | + |
| 15. | *Goryphus apollonis*\* | – | + | 46. | *Gryon* sp.\* | – | + |
| 16. | *Goryphus nursei*\* | + | + | 47. | *Hadronotus hogenakalense*\* | + | + |
| 17. | *Ichneumon* sp. | + | – | 48. | *Hadronotus fulviventris*\* | + | + |
| 18. | *Trathala flavoorbitalis* | + | + | 49. | *Protelenomus* sp. | – | + |
| 19. | *Temelucha* sp. | – | + | 50. | *Psix saccharicola* | – | + |
| 20. | *Xanthopimpla flavolineata* | + | + | 51. | *Scelio* sp. | + | + |
| 21. | *Xanthopimpla punctate* | – | + | 52. | *Telenomus dignus* | + | + |
| **Superfamily: Chalcidoidea** | | | | 53. | *Trimorus abhirupus*\* | – | + |
| **Family Agonidae** | | | | **Superfamily: Chrysidoidea** | | | |
| 22. | *Eupristina saundersi*\* | + | + | **Family Bethylidae** | | | |
| **Family Aphelinidae** | | | | 54. | *Goniozus indicus* | + | + |
| 23. | *Aphelinus asychis*\* | + | + | **Family Dryinidae** | | | |
| **Family Chalcididae** | | | | 55. | *Anteon achterbergi*\* | – | + |
| 24. | *Antrocephalus sepyra*\* | + | + | 56. | *Anteon yasumatsui*\* | – | + |
| 25. | *Brachymeria bengalensis* | + | + | 57. | *Aphelopus* sp. | + | + |
| 26. | *Brachymeria hime* | + | + | **Superfamily: Proctotrupoidea** | | | |
| 27. | *Brachymeria lasus* | + | – | **Family Diapriidae** | | | |
| 28. | *Brachymeria podagrica*\* | + | – | 58. | *Psillus* sp. | + | + |
| 29. | *Dirhinus auratus* | + | – | 59. | *Trichopria* sp. | + | – |
| 30. | *Kriechbaumerella kraussi*\* | + | + | **Superfamily: Ceraphronoidea** | | | |
| **Family Eulophidae** | | | | **Family Ceraphronidae** | | | |
| 31. | *Pediobius foveolatus*\* | + | + | 60. | *Aphanogmus fijiensis* | + | – |
| 32. | *Quadrastichus ovulorum*\* | – | + | | | | |

| S.No | Families/Parasitoids | 1st | 2nd | S.No | Families/Parasitoids | 1st | 2nd |
|------|----------------------|-----|-----|------|----------------------|-----|-----|
| **Family Eurytomidae** | | | | | | | |
| 33. | *Eurytoma* sp. | + | – | | | | |
| 34. | *Sycophila* sp. | | | | | | |
| **Family Mymaridae** | | | | | | | |
| 35. | *Anagrus atomus*** | + | – | | | | |

**Note:**
1st, first season; 2nd, second season;
* New record in Delhi;
** New recored in India; +, Presence; –, Absence

## DISCUSSION

Several studies carried out in different time-lines suggest the occurrence of diverse parasitoids on *L. orbonalis*. In addition to *T. flavoorbitalis*, several other Ichneumonids have been recorded including *Pristomerus testaceous* (*Ramakrishna Ayyar, 1927*), *Eriborus argentiopilosus* (*Tewari & Sardana, 1987*), *Xanthopimpla punctata* (*Navasero, 1983*; *Navasero & Calilung, 1990*), *E. sinicus* (*Talekar, 1995*) and *Diadegma apostate* (*Krishnamoorthy & Mani, 1998*). Several previous studies have reported the occurrence of braconids wasps *viz.*, *Chelonus* sp. (*Sandanayake & Edirisinghe, 1992*), *Bracon* sp. (*Tewari & Sardana, 1987*), *B. greeni* (*Venkatraman, Gupta & Negi, 1948*), and *Phanerotoma* sp. (*Tewari & Moorthy, 1984*; *Sandanayake & Edirisinghe, 1992*) *Apanteles* sp. (*Navasero & Calilung, 1990*). Among a diverse range of larval parasitoids, *T. flavoorbitalis* was recorded as the most critical species constituting about 60% of larval parasitoids. It has been recorded as a major parasitoid of *L. orbonalis* in Sri Lanka, Gujarat (India) and Bangladesh, with maximum parasitism of 61.70% (*Alam et al., 2003*). The superfamily Chalcidoidea species identified in relation to *L. orbonalis* *viz.*, *Brachymeria* sp., *B. obscurata* (*Navasero & Calilung, 1990*) *Brachymeria. lasus*, *Antrocephalus mitys* (Chalcididae), *S. irregularis*, *S. gemina, S. endius* (Spalangiidae) and *Trichogramma* sp. (Trichogrammatidae) (*Yasodha & Natarajan, 2009*; *Murali et al., 2017*).

In the present study, a total of five species belongs to five genera under three families (Braconidae, Ichneumonidae, and Spalangiidae). *T. flavoorbitalis* had the highest rate of parasitism in comparison to other parasitoids, demonstrating an average parasitism rate of 9.28%. In contrast, *B. greeni, A. hemara, S. gemina* and *G. nursei* displayed lower parasitism rates of 1.76%, 3.11%, 1.21%, and 1.06%, respectively. These findings are consistent with earlier studies conducted by *Alam et al. (2003)*, *Nagalingam (2006)*, and *Ranjith et al. (2020)*. *T. flavororbitalis* has been recorded as an important parasitoid in different countries, such as USA (*Swezey, 1926*), Bangladesh (*Alam & Sana, 1962*), Nepal (*Kafle, 1970*) and Sri Lanka (*Sandanayake & Edirisinghe, 1992*). It has been observed in different regions of India, Haryana (*Naresh, Malik & Balan, 1986*); Bihar (*Mallik et al., 1989*); Karnataka (*Ranjith et al., 2020*); Gujarat, Uttar Pradesh (*Alam et al., 2003*); and Manipur (*Thokchom et al., 2022*).

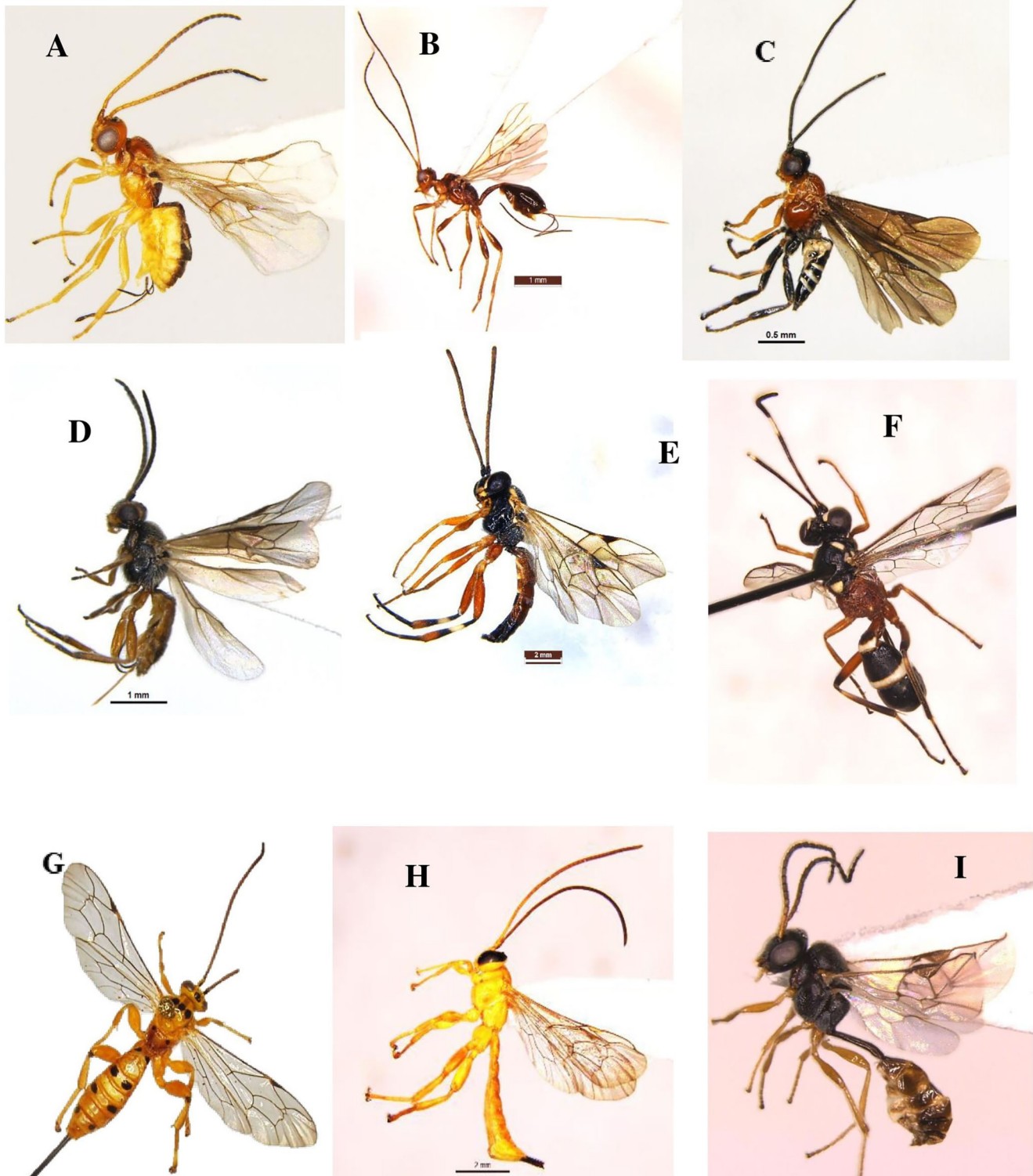

**Figure 8 Hymenoptera parasitoid species collected during the study period (November 2021 to October 2022).** (A) *Bracon carpomyiae* (*Ramakrishna Ayyar, 1927*); (B) *Spathius helle Nixon, 1965*; (C) *Amyosoma chinensis* (Szépligeti, 1902); (D) *Habrobracon* sp.; (E) *Diplazon laetatorius* (Fabricius, 1781); (F) *Goryphus apollonis* Jonathan & Gupta, 1973; (G) *Xanthopimpla punctata* (Fabricius, 1781); (H) *X. flavolineata* Cameron, 1907; (I) *Aneuclis* sp.

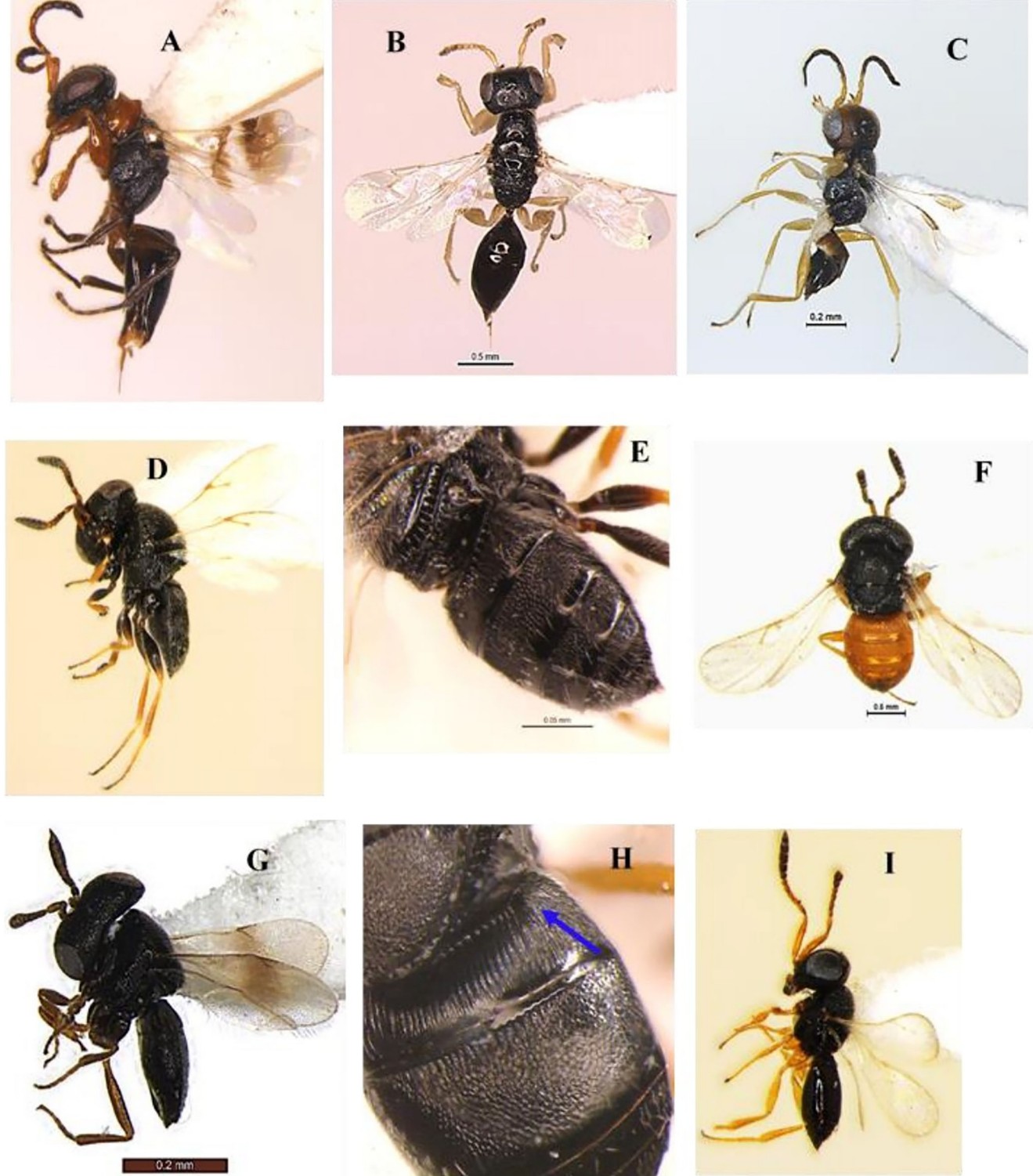

**Figure 9 Hymenoptera parasitoid species collected during the study period (November 2021 to October 2022).** (A) *Anteon achterbergi* Olmi, 1989; (B) *A. yasumatsui* Olmi, 1984; (C) *Aphelopus* sp.; (D and E) *Hadronotus hogenakalense* Sharma, 1982; (F) *H. fulviventris* (Crawford, 1912); (G and H) *Gryon* sp.; (I) *Telenomus dignus* (Gahan, 1925).

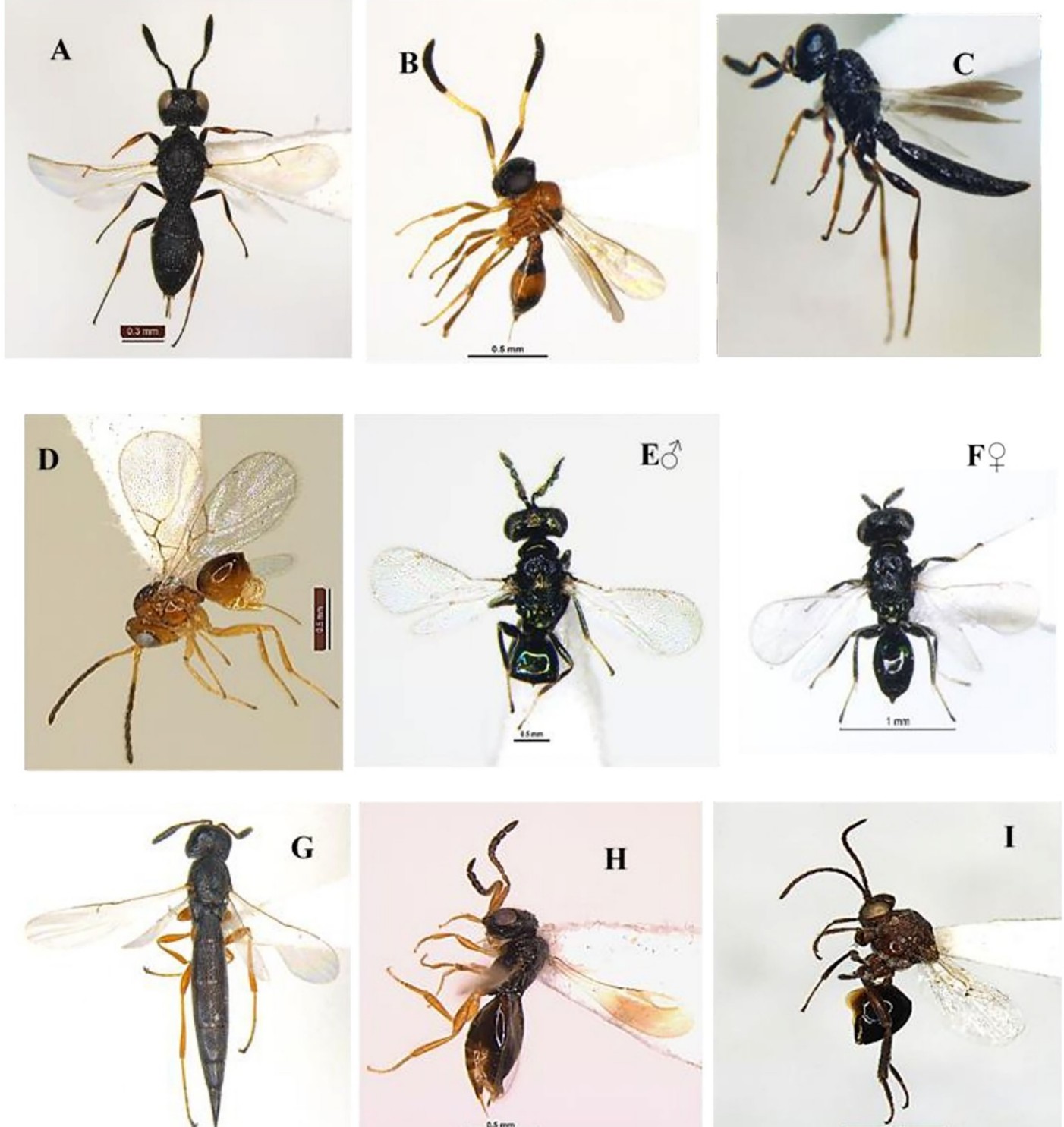

**Figure 10 Hymenoptera parasitoid species collected during the study period (November 2021 to October 2022).** (A) *Baryconus europaeus* (Kieffer, 1908); (B) *Trimorus abhirupus* Rajmohana & Sunita, 2023; (C) *Scelio* sp.; (D) *Alloxysta pleuralis* (Cameron, 1879); (E and F) *Pediobius foveolatus* (Crawford, 1912); (G) *Macroteleia livingstoni* Saraswat, 1982; (H) *Aphanogmus fijiensis* (Ferriere, 1933); (I) *Callaspidia notate* (Fonscolombe, 1832).

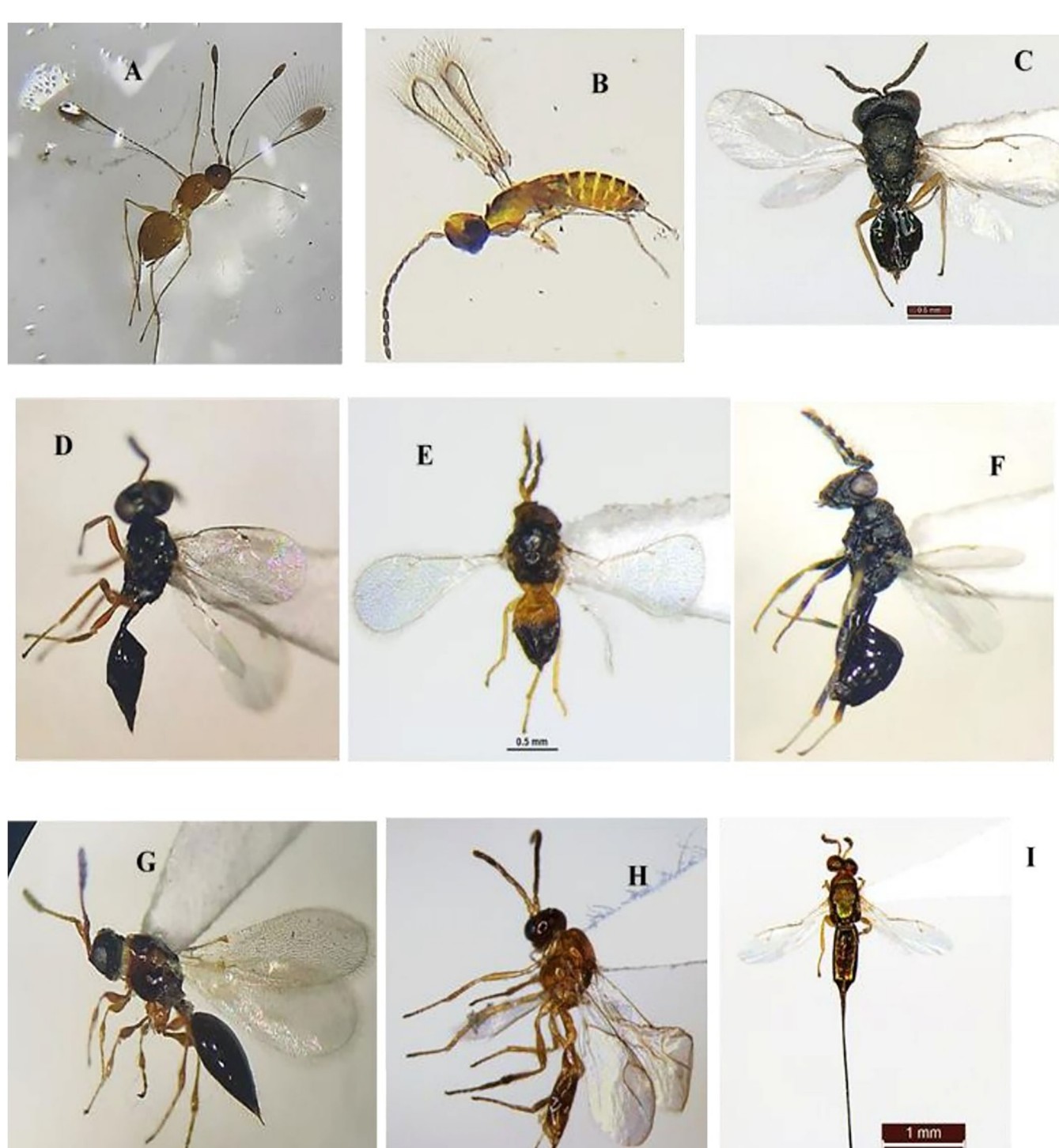

**Figure 11 Hymenoptera parasitoid species collected during the study period (November 2021 to October 2022).** (A) *Mymar taprobanicum* Ward, 1875; (B) *Anagrus atomus* (Linnaeus, 1767); (C) *Pachyneuron solitarium* (Hartig, 1838); (D) *Sphegigaster brunneicornis (Ferrière)*; (E) *Quadrastichus ovulorum* Ferriere, 1930; (F) *Eurytoma* sp.; (G) *Trichopria* sp.; (H) *Binodoxys indicus* (Subba Rao & Sharma, 1958); (I) *Sycoscapter* sp.

The current study revealed a higher maximum parasitism rate of 18.45% on larvae of *L. orbonalis* in August during 2nd season. This finding aligns with previous studies conducted by *Srinivasan (2008)* and *Ranjith et al. (2020)*, which indicate that the parasitoid potentially reduced the population of *L. orbonalis* (*Srinivasan, 2008*; *Kumar & Raghuraman, 2014*). This study recorded *A. hemara* as a parasitoid on *L. orbonalis* for the first time in Delhi. This finding aligns with the findings by *Navasero (1983)*, *Navasero & Calilung (1990)*, which recorded *Apanteles* sp. on BFSB in Philippines. According to *Alam et al. (2003)*, *G. nursei* was active in the winter but did not appear in the summer, suggesting a drop in its activity in response to elevated temperatures.

The brinjal fruit and shoot borer damage varied considerably, with the most substantial damage occurring in the summer season, while a minimal infestation of fruit was found during the winter season (Fig. 7). This could be attributed to elevated summer temperatures and subsequent cooling trends from December to January (Fig. 7). At the peak period, the pest damage exceeded 70% of total fruits, whereas a barely 27% damage was recorded in January. Generally, the level of parasitism concisely concurs with the population of the pest. However, a notable correlation was observed between the fruit infestation rate and the parasitism rate. These findings are consistent with the studies conducted by *Alam et al. (2003)* and *Ranjith et al. (2020)*.

## CONCLUSION

In summary, the fruit and shoot borer (BFSB) *L. orbonalis*, is a major pest of brinjal worldwide, causing extensive damage to the fruit and making it unsuitable for human consumption. A total of five parasitoid species were recorded on *L. orbonalis*, including the first-ever recorded instance of *Apanteles hemara* acting as a parasite on this pest in New Delhi. The survey revealed the presence of various Hymenoptera parasitoids within the brinjal crop ecosystem. However, the parasitism rate significantly varied depending on the environmental conditions during the survey. A significant positive correlation was observed between the parasitism rate with the fruit infestation rate. This study emphasizes the importance of preserving and protecting the natural enemies as they serve as effective biocontrol agents for *L. orbonalis*, thereby reducing the reliance on pesticides.

## LIST OF ABBREVIATIONS

| | |
|---|---|
| **T1** | First Metasoma tergite |
| **T2** | Second Metasoma tergite |
| **T3** | Third Metasoma tergite |
| **T8** | Eight Metasoma tergite |
| **1−R1** | Metacarp of the fore wing |
| **r** | First radial abscissa |
| **2Rs** | First cubital cross vein |
| **3-SR** | Second abscissa of radius |
| **ICAR** | Indian Council of Agricultural Research |
| **HC** | Hager collection |

### Funding

The ICAR-Indian Agricultural Research Institute, Division of Entomology, provided necessary research facilities. This work was supported by Princess Nourah bint Abdulrahman University, Riyadh, Saudi Arabia, through the University Researchers Supporting Project (no. PNURSP2023R37). The funders had no role in study design, data collection and analysis, decision to publish, or preparation of the manuscript.

### Grant Disclosures

The following grant information was disclosed by the authors:
The ICAR-Indian Agricultural Research Institute.
Division of Entomology.
Abdulrahman University, Riyadh, Saudi Arabia, through the University Researchers Supporting Project: PNURSP2023R37.

### Competing Interests

The authors declare that they have no competing interests.

### Author Contributions

- Hager M. M. Saleh conceived and designed the experiments, performed the experiments, analyzed the data, prepared figures and/or tables, authored or reviewed drafts of the article, and approved the final draft.
- Areej A. Al-Khalaf conceived and designed the experiments, prepared figures and/or tables, authored or reviewed drafts of the article, and approved the final draft.
- Maha Abdullah Alwaili performed the experiments, prepared figures and/or tables, authored or reviewed drafts of the article, and approved the final draft.
- Debjani Dey conceived and designed the experiments, performed the experiments, analyzed the data, prepared figures and/or tables, authored or reviewed drafts of the article, and approved the final draft.

### Data Availability

The raw data are available in the Supplemental File.

### Supplemental Information

Supplemental information for this article can be found online at http://dx.doi.org/10.7717/peerj.16870#supplemental-information.

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
