# Peer review of "First record of Apanteles hemara (N.) on Leucinodes orbonalis Guenée and biodiversity of Hymenoptera parasitoids on Brinjal"

_PeerJ, doi:10.7717/peerj.16870_

## Round 0.1 · original submission · Major Revisions

Dear Authors

The manuscript cannot be accepted for publication in its current form. It needs a major revision to be reconsidered for publication. The authors are invited to revise the paper considering all the suggestions made by the reviewers. Please note that requested changes are required for publication.

With Thanks

Reviewer 1 ·

Basic reporting

The research was based on a good period of work, and the study included enough references to a large extent, and a good number of parasitoids were collected from the locality of the study.
Therefore, the study is considered adequate based on what the authors specified.

Experimental design

No comment

but the author should include "list of abbreviations"

Validity of the findings

GOOD

The definition of species seems correct depending on the references listed by the authors, and there is no problem with it, but in identifying parasitoids there is usually a lot of overlap in the characteristics, and therefore it is always advisable to make sure of the definition of the good side.

Although the authors were satisfied with the diagnosis for each species
But it is preferable to add some diagnositic morphometric characters.

some latin names need to be italicized (example: line 204 and 229)

Additional comments

No comment

Reviewer 2 ·

Basic reporting

Dear Editor and Authors,

I am writing to provide a comprehensive review of the manuscript titled "First record of Apanteles hemara (N.) on Leucinodes orbonalis Guenée and biodiversity of Hymenoptera parasitoids on Brinjal" submitted to your esteemed journal. While the study holds promise and contributes to the field, there are several critical issues that require attention and improvement before it can be considered suitable for publication.

Image Quality and Presentation:
The images included in the manuscript are of notably poor quality and appear to be inadequately integrated into the paper. The overall image quality severely detracts from the scientific value of the paper. I strongly recommend that the author revisit this aspect and consider retaking the photographs, ensuring they meet the necessary standards for clarity and relevance. High-quality visuals are essential for conveying the research effectively.

Graphical Representation:
The graphical elements, such as graphs and figures, also suffer from a lack of quality. I urge the authors to remake these graphical representations to ensure clarity and accuracy. Clear, well-constructed visuals are vital for conveying complex scientific information.

Language Quality:
The manuscript exhibits issues with the English language, which hinders the reader's comprehension. I recommend that the authors seek assistance from a fluent English speaker or professional language editor to enhance the overall clarity and readability of the manuscript. A well-written paper is essential for effective communication in the academic community.

Statistical Analysis:
The statistical analysis employed in this study does not appear to be valid. I strongly encourage the authors to consider utilizing the R statistical software, which is a validated and widely accepted tool for statistical analysis in scientific research. This will enhance the credibility and rigour of their findings.

Species Diagnosis:
The paper includes a species diagnosis that appears to be largely repetitive of existing research. It is essential that the authors base their diagnosis on their own specimens, rather than relying on those of other researchers. Failure to do so not only raises ethical concerns but also reflects negatively on their academic credibility and the reputation of the journal.

IMRD Sections:
The various sections of the manuscript, following the IMRD (Introduction, Methods, Results, Discussion) structure, are insufficient both in technical content and linguistic quality. Each section should be improved to provide a more comprehensive and well-structured account of the research.

I kindly request that the authors consider these recommendations and revise their manuscript accordingly. Once these concerns have been adequately addressed, I believe the paper will make a valuable contribution to the scientific community.

Thank you for considering my review.

Sincerely,

Experimental design

As previously stated.

Validity of the findings

As previously stated.

Additional comments

As previously stated.

·

Basic reporting

I would not say these are "failures" just a couple of instances where taxonomy and citations need to be updated.
1. The change in family from Pteromalidae to Spalangiidae. The specific reference for this change is:
Burks R, Mitroiu M-D, Fusu L, Heraty JM, Janšta P, Heydon S, Papilloud ND-S, Peters RS, Tselikh EV, Woolley JB, van Noort S, Baur H, Cruaud A, Darling C, Haas M, Hanson P, Krogmann L, Rasplus J-Y (2022) From hell’s heart I stab at thee! A determined approach towards a monophyletic Pteromalidae and reclassification of Chalcidoidea (Hymenoptera). Journal of Hymenoptera Research 94: 13-88. https://doi.org/10.3897/jhr.94.94263

2. A new version of the Universal Chalcidoidea Database, with a new method of of citation. The new web address for the database is: https://ucd.chalcid.org/#/

Experimental design

Regarding sufficiency of methodological detail, It's not clear to me that Spalangia gemina was a primary parasitoid of L. orbonalis, and not acting as a hyperparasitoid on one or more of the Ichneumonoidea. The species is known to act as a hyperparasitoid on Braconidae. Is there evidence from this study that S. gemina is acting as a primary parasitoid, or are the authors relying on Murali et al., (2017) for the host association?

Validity of the findings

The final sentence of the conclusion extrapolates a bit beyond the data. No data data are presented that suggest that augmentative or conservation biological control can reduce pesticide use in this system. Controlled studies examining crop yield under different pest management regimens would be needed to determine the economic viability of reducing pesticide use.

·

Basic reporting

This manuscript has been written with a good level of English, it is clear and technically accurate. The literature used is sufficient, and the study's background has been presented satisfactorily.
All the figures and tables appear to complement the manuscript well; I do not see a need to remove them. The results are provided straightforwardly, and the discussion aligns with the stated objectives.

Experimental design

The paper is straightforward; it's not a groundbreaking or immensely novel research proposal. However, the journal is more interested in direct, relevant, and practical submissions - qualities that are indeed present in this proposal.

The methods employed are adequate, but some additional details are needed to facilitate the replication of this work (more details in “General comments section”).

Validity of the findings

This is a somewhat novel, localized study with the potential for replication and citation. Additionally, it provides tools for future research, especially considering applied science for agricultural pest control.

The results obtained are discussed precisely and concisely. While the conclusion is somewhat brief, it effectively serves its purpose by being direct and addressing the fulfillment of the proposed objectives.

Additional comments

This proposal presents a study on the brinjal fruit and shoot borer (BFSB) and its parasitoids, which is undoubtedly a significant topic given the economic losses it causes to the brinjal crop globally. However, the manuscript has several minor and major issues that call for a major revision.
While the study's objective is clear, some aspects need improvement. Firstly, the methodology and experimental setup should be described more comprehensively to enhance the research's reproducibility and clarity. Additionally, the presentation of results lacks context and detail.
In conclusion, the text highlights a relevant study on BFSB and its parasitoids, but it requires substantial refinement to provide a clearer understanding of the research methodology and results.

Below, I outline some general aspects that require revision:

According to the author guidelines, when there are three authors, their surnames should be cited as per the instructions below:
'In-text citations • For three or fewer authors, list all author names (e.g., Smith, Jones & Johnson, 2004). For four or more, abbreviate with 'first author' et al. (e.g., Smith et al., 2005).' Review all references with three authors.
Review when citing books: 'Example book reference: James FY. 2010. Understanding corn and wheat. Oxford: Oxford University Press.' Review the books mentioned in the references section.
The caption for Figure 2 needs to be revised (comment in the original document). Figure 8 needs to be revised (comment in the original document).
Provide more detail on the data collection methodology.
The taxonomical study lacks information about insect families. Additionally, two species names are not in italics.

Everything that needs to be reviewed or corrected, along with my comments, suggestions, and corrections, will be provided directly in the manuscript file to streamline the revision process for the authors. I've also made some improvements for clarity and conciseness.

Considering this proposal to be intriguing and having the potential for publication in PeerJ, my suggestion is to accept the publication after a more comprehensive revision.

---

## Round 0.2 · Minor Revisions

Dear Authors

The manuscript still needs a minor revision to be reconsidered for publication. The authors are invited to revise the paper considering all the suggestions made by the reviewers. Please note that requested changes are required for publication.
With Thanks

Reviewer 2 ·

Basic reporting

Refer to my additional comments

Experimental design

Refer to my additional comments

Validity of the findings

Refer to my additional comments

Additional comments

Many of the earlier comments remain unattended. Consequently, I'll extend another opportunity to the authors to revise their manuscript. Failure to address these concerns may result in the manuscript being rejected.

·

Basic reporting

There are still minor issues with English throughout, but these are mostly stylistically awkward rather than technically incorrect. There are no English issues that impede understanding. I've made few suggestions below:
Line 28: spell out genus names when they start a new sentence (change “T.” to “Trathala”).
Line 105: delete the second instance of “weekly”
Line 110: change “parasites” to “parasitoids”
Line 111: change “parasites” to “parasitoids”, and change “big parasitoids” to “large specimens (approx. #mm or larger)”. Replace my number symbol with the approximate minimum length of a specimen you considered to be large.
Lines 131-133: Change to “The findings of this study revealed the presence of five parasitoid species on L. orbonalis: A. hemara and B. greeni (Braconidae); G. nursei and T. flavoorbitalis (Ichneumonidae); and S. gemina (Spalangiidae).” Why are family names given f
Line 279: delete “on the data obtained under this study”
Line 329: reword; Hawaii is a U.S. state, not a country.

Experimental design

In my initial review, I asked for evidence from this study indicating that S. gemina was acting as a primary parasitoid of the pest, and not as a hyperparasitoid on one of the ichneumonoids. The authors reply is to state that it "appears" as a primary parasitoid, then cite other literature indicating that it can act as both a primary and hyperparasitoid. "Appears" as a primary parasitoid is not evidence.

Validity of the findings

no comment

Additional comments

no comment

·

Basic reporting

The authors have taken into account all of my comments and responded to each comment with great attention and precision. I am satisfied with the modifications made and the responses provided. I recommend accepting the publication.

Experimental design

The authors have taken into account all of my comments and responded to each comment with great attention and precision. I am satisfied with the modifications made and the responses provided. I recommend accepting the publication.

Validity of the findings

The authors have taken into account all of my comments and responded to each comment with great attention and precision. I am satisfied with the modifications made and the responses provided. I recommend accepting the publication.

Additional comments

The authors have taken into account all of my comments and responded to each comment with great attention and precision. I am satisfied with the modifications made and the responses provided. I recommend accepting the publication.

---

## Round 0.3 · accepted · Accept

Dear Authors,
I am pleased to inform you that after the last round of revision, the manuscript has been improved a lot, and it can be accepted for publication.
Congratulations on accepting your manuscript, and thank you for your interest in submitting your work to PeerJ.
With Thanks

Reviewer 2 ·

Basic reporting

Unfortunately, the authors overlooked most of the comments. I provided them with two chances to improve their manuscripts, but I haven't witnessed significant progress. Therefore, I just rejected it.

Experimental design

NA

Validity of the findings

NA

Additional comments

NA